# Whole-Cell Biocatalytic Production of Acetoin with an *aldC*-Overexpressing *Lactococcus lactis* Using Soybean as Substrate

**DOI:** 10.3390/foods12061317

**Published:** 2023-03-20

**Authors:** Huajun Luo, Weihong Liu, Yiyong Luo, Zongcai Tu, Biqin Liu, Juan Yang

**Affiliations:** 1National R&D Center for Freshwater Fish Processing, College of Life Sciences, Jiangxi Normal University, Nanchang 330022, China; 2Faculty of Life Science and Technology, Kunming University of Science and Technology, Kunming 650500, China

**Keywords:** acetoin biosynthesis, whole-cell biocatalyst, soybean, optimization, α-acetolactate decarboxylase, fermentation, heterologous expression

## Abstract

Douchi is a traditional Chinese fermented soybean product, in which acetoin is a key flavor substance. Here, the α-acetolactate decarboxylase gene *aldC* was cloned from *Lactiplantibacillus* (*L.*) *plantarum* and overexpressed in *Lactococcus* (*L.*) *lactis* NZ9000 by nisin induction. The ALDC crude enzyme solution produced an enzyme activity of 35.16 mU. Next, whole cells of the recombinant strain NZ9000/pNZ8048-*aldC* were employed as the catalyst to produce acetoin in GM17 medium. An optimization experiment showed that an initial OD_600_ of 0.6, initial pH of 7.5, nisin concentration of 20 ng/mL, induction temperature of 37 °C and static induction for 8 h were the optimal induction conditions, generating the maximum acetoin production (106.93 mg/L). Finally, after incubation under the optimal induction conditions, NZ9000/pNZ8048-*aldC* was used for whole-cell biocatalytic acetoin production, using soybean as the substrate. The maximum acetoin yield was 79.43 mg/L. To our knowledge, this is the first study in which the *aldC* gene is overexpressed in *L. lactis* and whole cells of the recombinant *L. lactis* are used as a biocatalyst to produce acetoin in soybean. Thus, our study provides a theoretical basis for the preparation of fermented foods containing high levels of acetoin and the biosynthesis of acetoin in food materials.

## 1. Introduction

Acetoin (i.e., 3-hydroxy-2-butanone) is a compound with a molecular formula of C_4_H_8_O_2_ (CAS#513-86-0) and a relative molecular weight (MW) of 88.11. As chiral molecules, the (3S)- and (3R)-enantiomers are important potential pharmaceutical intermediates [1]. Due to its pleasant yogurt-like scent and creamy butter taste, acetoin is widely used in foods and cosmetics [2]. According to the Joint FAO/WHO Expert Committee on Food Additives (JECFA) and the Food and Drug Administration (FDA), acetoin is a generally recognized as safe (GRAS) substance. Traditional chemical synthesis is used to produce most of the current commercial acetoin, which is unsafe to use in foods and causes serious environmental pollution. In contrast, the biological processes for acetoin production have attracted extensive attention due to their low production cost and environmental friendliness. Recently, some studies reported that natural or engineered microorganisms were used in the biotechnological production of acetoin [3,4,5].

Acetoin is naturally produced by microorganisms and other species, and compared to eukaryotes, bacteria can produce acetoin much more efficiently [1]. Typically, acetoin can be biosynthesized in the following steps: (1) pyruvate forms α-acetolactate by α-acetolactate synthetase based on pyruvate decarboxylation catalyzed with the help of thiamin diphosphate-dependent enzymes; (2) α-acetolactate is converted to acetoin by α-acetolactate decarboxylase (ALDC). Alternatively, α-acetolactate can be converted to diacetyl by a nonenzymatic oxidative decarboxylation reaction, and further reduced to acetoin by a diacetyl reductase [1]. In fact, acetoin is not the final compound, as it is usually reduced to 2,3-butanediol by 2,3-butanediol dehydrogenases [6]. ALDC forms physiologically relevant dimers, in which each monomer contains two domains with an α/β tertiary structure [7]. Further structure analysis shows that three histidine residues (H191, H193 and H204) from the β13 and β14 strands accommodate a zinc ion, suggesting that the zinc ion is an important metal ion in acetolactate decarboxylation [7]. ALDC is often considered the rate-limiting key enzyme in the acetoin/2,3-butanediol biosynthesis pathway [8,9]. The nisin-controlled gene expression system (NICE) was one of most frequently used gene expression systems in Gram-positive bacteria. This system was adopted for the “food-grade” production of recombinant proteins, and nisin is an efficient food compound inducer.

Biotransformation is a process of chemical modification in which a biocatalyst converts a raw chemical compound into a structurally related high-value-added product, and this process is often mediated by purified enzyme and whole cells [10]. Traditional chemical synthesis exhibits disadvantages over biotransformation, such as low specificity, harsh conditions, complex purification steps, and byproduct formation [11]. The enzyme-containing cells were harvested and used in the whole-cell biotransformation, which can eliminate the separation and purification process of the enzyme, thus reducing production costs. Under the protection of the cell membrane, the structures of enzymes were more stable and the enzymes were more efficient [12]. Whole-cell biotransformation can also use cofactors that are already produced by the cells [11]. Due to its advantages, including mild reaction conditions, high titers, and ease of industrial and environmental protection, whole-cell bioconversion has become an attractive method of chemical production.

In recent years, the metabolic engineering of acetoin-producing microorganisms, such as *Escherichia* (*E*.) *coli* [3] and *Klebsiella* (*K*.) *pneumoniae* [4], has been used to improve the yield and productivity of acetoin. However, as opportunistic pathogens, *E. coli* and *K. pneumoniae* cannot be used in food and cosmetics. *Lactococcus* (*L.*) *lactis*, a typical lactic acid bacterium, is a GRAS microorganism and was demonstrated to be a native acetoin producer [2]. Hence, in this study, the *aldC* gene was cloned from *Lactiplantibacillus* (*L.*) *plantarum* and overexpressed in *L. lactis*. Different conditions for *aldC* gene expression were analyzed to find the optimal conditions for acetoin production. Under the optimal conditions, the recombinant strain was collected and soybeans were used as substrates for the whole-cell biotransformation production of acetoin. To our knowledge, this is the first time that the *aldC*-overexpressing *L. lactis* strain as a whole-cell biocatalyst was used to produce acetoin with soybean. This study used soybean as a substrate to produce acetoin, providing a theoretical basis for the production of high-value products from natural food materials.

## 2. Materials and Methods

### 2.1. Strains, Media and Regents

The bacterial strains used in this study are listed in Table 1. *L. plantarum* Ly8 was isolated from Douchi (a traditional Chinese salt-fermented soybean) and grown at 37 °C in de Man, Rogosa and Sharpe (MRS) medium (HKM, Guangdong, China). *L. lactis* strain NZ9000 (hereafter referred to as NZ9000) was cultivated in M17 Broth (Sangon, Shanghai, China) with 0.5% glucose (GM17) at 30 °C. *E. coli* DH5α, as the host cell for plasmid construction, was cultured in Luria Broth (LB) liquid medium or plated on LB agar at 37 °C. When needed, chloramphenicol (OXOID, Hants, UK) was used as a selection agent. As a precursor of acetoin, α-acetolactate was chemically prepared as follows: ethyl-2-acetoxy-2-methylacetoacetate (Sigma-Aldrich, St. Louis, MO, USA), 1 M NaOH and sterilized dH_2_O were mixed at a ratio of 55:40:5 (*v*/*v*/*v*), and the mixture was reacted in an ice-water bath for 15 min. Because α-acetolactate is chemically unstable and can be converted to diacetyl without any enzyme [13], it must be used immediately after it is ready. A chromogenic agent for acetoin determination is also ready-to-use, and is generated as follows: 0.1 g of creatine and 1 g of α-naphthol were dissolved in 100 mL of 1 M NaOH.

### 2.2. Gene Cloning, Sequence Analysis and Homology Modeling

Genomic DNA of *L. plantarum* Ly8 was extracted using a DNAprep Pure Bacteria Kit (Bioteke, Beijing, China). An *aldC* gene was amplified using the primers *aldC*-F and *aldC*-R (Table 1), and the PCR product was directly sequenced. The amino acid sequence and its theoretical isoelectric point and MW were deduced using EditSeq software 7.1.0 (DNASTAR Inc., Madison, WI, USA). The deduced amino acid sequence was submitted to the NCBI GenBank database (http://www.ncbi.nlm.nih.gov/BLAST accessed on 6 December 2022), and its homologs from different bacteria were downloaded. A maximum likelihood tree was constructed using the MEGA X software package (Mega Limited, Auckland, New Zealand). The phylogenetic tree generated was manually edited for improved visualization using the iTOL web tool (https://itol.embl.de accessed on 3 December 2022). The GenBank accession number of ALDC is OQ082579. The putative secondary structures were predicted based on homology modeling, with the ALDC protein structure of *K. pneumoniae* (PDB code 6INC, https://www.rcsb.org/structure/6INC accessed on 6 December 2022) as a template [14] using the ESPript/ENDscript 3.0 software package (https://espript.ibcp.fr/ESPript/cgi-bin/ESPript.cgi accessed on 6 December 2022).

### 2.3. Plasmid Construction and Gene Expression in L. lactis

The gel-purified *aldC* gene fragment was digested with *Pst* I and *Xba* I to obtain products with sticky ends, which were then cloned into the pNZ8048 vector (Table 1) digested with the same restriction enzymes. Putative positive transformants were selected by plating on LB agar containing 50 μg/mL chloramphenicol and verified by colony PCR. Then, the recombinant plasmid (pNZ8048-*aldC*) was electrotransformed (25 μF capacitance, 200 Ω resistance, 10 kV/cm and 4.5 ms) into the host strain NZ9000, and positive transformants were selected on GM17 agar medium containing 10 μg/mL chloramphenicol. The host strain transformed with an empty vector was used as the control (NZ9000/pNZ8048). For gene expression, recombinant NZ9000 harboring pNZ8048-*aldC* (NZ9000/pNZ8048-*aldC*) was statically incubated at 30 °C in GM17 medium with chloramphenicol (10 μg/mL) to an OD_600_ of 0.4, and then 10 ng/mL nisin was added. After further incubation for 4 h, the cells were harvested by centrifugation at 10,000 rpm for 10 min at 4 °C, washed, and suspended in 15 mL phosphate-buffered saline (PBS) buffer (pH 7.4). Then, the cells were disrupted by sonication (Φ3, 50%, on for 6 s and off for 4 s, for a total of 4 min) on ice. The pellet and supernatant were collected by centrifugation at 12,000 rpm for 15 min and analyzed by SDS-PAGE to test whether ALDC was expressed. SDS-PAGE was performed in accordance with Mohseni’s method [15].

### 2.4. Determination of Enzyme Activity

Two samples of recombinant bacterium NZ9000/pNZ8048-*aldC* were cultured, harvested and disrupted as described above, except that one sample was induced by 10 ng/mL nisin and the other was not. The supernatant was collected by centrifugation to obtain a crude enzyme solution, and its enzymatic activity was assayed by measuring the production of acetoin. One unit of enzyme activity is defined as the formation of 1 μmol of acetoin per minute [16]. Briefly, 200 μL of crude enzyme solution and 200 μL of freshly prepared α-acetolactate solution were mixed and reacted in a water bath at 30 °C for 20 min. The acetoin produced in 400 μL of solution reacted with 4.6 mL of chromogenic agent at room temperature for 50 min to develop a pink color, and its concentration was measured at 522 nm with acetoin as a standard. The acetoin standard curve was plotted as the optical density (OD_522_) vs. the acetoin concentration. Parallel experiments were performed with 200 μL of MES buffer (0.05 M 4-morpholineethanesulfonic acid, 0.05% Brij-35, pH 6) instead of crude enzyme solution, and the resulting OD_522_ value was used for zeroing.

### 2.5. Optimizing the Concentration of Nisin

To select an appropriate nisin concentration, 0–40 ng/mL nisin was added to the culture when the OD_600_ of NZ9000/pNZ8048-*aldC* reached 0.4. After 4 h of induction at 30 °C, cells were collected and disrupted as described above, and nisin-induced ALDC in the supernatant was detected by Western blotting (WB). Briefly, total proteins were separated on SDS-PAGE gels and then transferred to polyvinylidene fluoride (PVDF) membranes by using an electrophoretic transfer system. The membrane was blocked in 5% bovine albumin serum (Sigma-Aldrich) and incubated overnight at 4 °C with anti-6× His Tag antibody (ABclonal, Wuhan, China; 1:5000 dilution). The membrane was washed in TBST buffer (20 mM Tris-HCl, 136.7 mM NaCl, 0.5% Tween 20, pH 7.5) five times and then blotted in HRP goat-anti-mouse IgG (H + L) antibody (ABclonal, Wuhan, China; 1:10,000 dilution) at room temperature for 1 h. Then, the membrane was washed again with TBST buffer, and the protein bands were visualized using ECL (enhanced chemiluminescence) Plus reagents (Beyotime Biotech Inc., Shanghai, China). A ChemiDoc XRS imaging system (Bio-Rad, Hercules, CA, USA) was used to detect the bands after the immune response.

### 2.6. Optimizing the Substrate/Biomass Ratio in a Whole-Cell Bioconversion System

An appropriate ratio of substrate (α-acetolactate) to bacterial biomass is important for accurately measuring the acetoin production capacity of recombinant bacteria in a whole-cell biotransformation system. NZ9000/pNZ8048-*aldC* was incubated statically at 30 °C until the OD_600_ reached 0.4 and divided into 10 mL/tube. Nisin was added into each tube at a final concentration of 20 ng/mL and incubated at 30 °C for 4 h. The cells were collected by centrifugation (10,000 rpm, 10 min, 4 °C), washed twice with PBS buffer, and resuspended in 24 mL PBS as the whole-cell biocatalyst to produce acetoin. Then, 1/4, 1/8, 1/16, 1/32, 1/64 and 1/128 of the whole-cell biocatalyst was harvested by centrifugation. Each cell sediment was mixed with 3 mL α-acetolactate solution, prepared as mentioned above, and reacted for 60 min at 30 °C to produce acetoin. Finally, the concentration of acetoin was calculated by the chromogenic reaction to determine the optimal ratio of substrate and biomass.

### 2.7. Optimizing the Induction Conditions

NZ9000/pNZ8048-*aldC* was inoculated in GM17 medium containing 10 μg/mL chloramphenicol, and the optimal nisin concentration revealed by the above experiment was used to induce ALDC expression. Other induction parameters, including initial OD_600_ (0.2, 0.4, 0.6, 0.8, and 1.0), initial pH (5.5, 6.0, 6.5, 7.0, 7.5, and 8.0), induction temperature (28, 30, 35, 37, and 40 °C), shaking speeds (0, 50, 100, and 150 rpm), and induction time (2, 4, 6, 8, and 10 h), were optimized to improve ALDC expression and/or activity, and the results were evaluated by a chromogenic reaction in an optimized whole-cell biotransformation system. To further increase acetoin production, an L_18_ (3^5^) orthogonal test (Table 2) was performed to analyze the optimal combination of induction conditions.

### 2.8. Whole-Cell Biosynthesis of Acetoin Using Soybeans as Substrate

Soybeans were pretreated according to our previous study [17]. NZ9000/pNZ8048-*aldC* was incubated and expressed ALDC under the optimal conditions revealed by the above experiments. Cells were harvested by centrifugation, washed and resuspended in PBS buffer to an OD_600_ of 1.4. Then, 2.5 mL of this whole-cell catalyst was added to 60 g of the cooled pretreated soybeans, and a biotransformation process was first performed at 37 or 40 °C for 24 h and then at 4 °C for 24 h. Finally, 10 g of the fermented soybeans was ground well, and metabolites were extracted twice with 20 mL of deionized water (10 mL each) by ultrasonication (200 W, 53 kHz, 30 min). After centrifugation (10,000 rpm, 10 min), the supernatant was collected and fixed to 25 mL with deionized water and acetoin in 400 μL supernatant was determined by the chromogenic reaction as described above. The whole acetoin biosynthetic process can be artificially divided into phase I (i.e., bacterial growth in MRS medium) and phase II (i.e., bacterial growth in soybeans). For comparative analysis, the fermented soybeans (FS) were collected from (1) nisin-containing phase I and nisin-free phase II, (2) nisin-containing phases I and II, and (3) nisin-free phases I and II, and were named FS1, FS2, and FS3, respectively. In addition, a PBS buffer containing 20 ng/mL nisin was added to the pretreated soybeans instead of the whole-cell catalyst, and then a parallel experiment was performed as described above and the resulting sample was considered soybean control (SC).

### 2.9. Statistical Analysis

Each assay was conducted at least in triplicate. Data were subjected to Student’s t test for two sets of data or one-way ANOVA for multiple sets of data, and a value of *p* < 0.05 was considered significant. All statistical analyses were conducted using SPSS 22 (IBM, Armonk, NY, USA).

## 3. Results and Discussion

### 3.1. Cloning, Sequence and Phylogenetic Analysis of the aldC Gene

The predicted open reading frame (ORF) of the *aldC* gene is 711 bp, encoding a polypeptide of 236 amino acid residues. The theoretical MW and isoelectric point of ALDC are 25.96 kDa and 5.07, respectively. The amino acid alignment results showed that ALDC from *L. plantarum* Ly8 was 100% identical to ALDC from *L. plantarum* WCFS1 (Figure 1). Five α-helices, fifteen β-sheets, four β-turns and two 3/10-helices were predicted in the secondary structures of the protein, which was similar to the ALDC crystal structure reported previously [7]. Based on the complete polypeptide sequences of ALDC from *L. plantarum* Ly8 and other bacteria, we built a phylogenetic tree (Figure 2) and found that ALDC from *L. plantarum* Ly8 was most closely related to *L. plantarum* ALDC. In the entire phylogenetic tree, ALDC sequences are clustered into several large clades depending on the species, which suggested that ALDC had evolved several times between different species long ago.

### 3.2. Heterologous Expression and Enzyme Activity Analysis

As shown in Figure 3a, an intense protein band close to 29 kDa was seen for the cell pellet and supernatant, in agreement with the expected MW of 25.96 kDa, which was consistent with a previous report by Eom et al. [7], suggesting that the ALDC from *L. plantarum* Ly8 was successfully expressed in soluble form in NZ9000; this result was further confirmed by the following WB experiment. To explore the activity of ALDC, acetoin was synthesized in vitro using α-acetolactate as a substrate and assessed by the chromogenic reaction (Figure 3b). Acetoin production was calculated by combining the acetoin standard curve and the optical density value at 522 nm (Figure 3c). Nisin-induced acetoin production was 8.87-fold (154.91 mg/L) higher than that without nisin induction (Figure 3d). Coincidently, nisin induction increased ALDC activity from 4.00 mU to 35.16 mU (Figure 3e). These results indicated that the expressed ALDC can efficiently convert α-acetolactate to acetoin. Notably, we detected minimal basal ALDC expression in the absence of nisin induction. The mechanism of plasmid pNZ8048 expression in induction absence is not clear, but some researchers have reported that other inducers, such as lactose and galactose, have the ability to induce this promoter at low levels [18].

### 3.3. Optimizing the Concentration of Nisin to Increase ALDC Expression

The results of the WB experiment showed that the expression of ALDC first increased and then decreased with increasing nisin intensity, and the maximum expression appeared when the concentration of nisin was 20 ng/mL (Figure 4). The effect of nisin concentration on *L. lactis* recombinant protein expression has been discussed in many studies. Bahey et al. [19] reported that 0.2% (*v*/*v*) nisin (filter-sterilized culture supernatant of the nisin-secreting strain *L. lactis* NZ9700 was used as a source of nisin) was chosen for 4 h induction to maximize Listeriolysin O production. Van Hoang et al. [20] reported that the maximal expression of recombinant T-cell epitope peptide was achieved with 40 ng/mL nisin induction for 4 h at 30 °C. In this study, 20 ng/mL nisin was selected for subsequent experiments.

### 3.4. Optimization of Substrate/Biomass Ratio and Induction Conditions

To fully evaluate the whole-cell biotransformation effect of NZ9000/pNZ8048-*aldC*, it is necessary to ensure that sufficient substrate (α-acetolactate) is available. As shown in Figure 5a, the production of acetoin did not change significantly when the biomass was diluted four times, and then the absorbance value at 522 nm began to decrease after the biomass was further diluted, meaning that α-acetolactate is sufficient from this point onward. Therefore, we finally chose to dilute the biomass 16 times for subsequent experiments to ensure sufficient substrate. Next, the induction conditions for ALDC expression and/or activity were optimized to increase acetoin production. As a result, all five single variables increased and then decreased the production of acetoin, and an initial OD_600_ of 0.6, pH of 6.5, induction temperature of 35 °C, shaking speed of 50 rpm and induction time of 6 h resulted in the maximum production of 94.47, 95.53, 103.51, 102.01, and 96.44 mg/L, respectively (Figure 5b–f). In terms of the induction temperature, a 1.77% reduction in acetoin was obtained at 37 °C compared to 35 °C (Figure 5e). The difference in the optimal temperature was attributed to interspecific specificity. For instance, ALDC from a mesophilic origin of *Streptococcus thermophilus* presented the highest enzyme activity, at 50 °C [21]. Shaking speed is an important factor in microbial metabolism. Too high of a speed causes cells to overgrow, undergo autolysis, and produce harmful byproducts, while a low-speed causes a lack of ATP and inhibits growth [22]. Considering that the host strain NZ9000 is a facultative anaerobe, a relatively low rotation speed (e.g., 50 rpm) favors cell growth and proper protein folding, which is consistent with the research of Li et al. [8]. For the initial pH, the preference of ALDC for weakly acidic conditions is similar to that of the other ALDCs reported earlier [21].

### 3.5. Orthogonal Test of the Induction Conditions for Improving Acetoin Production

The initial OD_600_ (R_X1_), initial pH (R_X2_), induction temperature (R_X3_), shaking speed (R_X4_) and induction time (R_X5_), which were obtained by the previous single-factor experiment, showed a significant influence on acetoin production. L_18_ (3^5^) was designed by an orthogonal test, with the best data as three levels of each factor. It can be seen from Table 3 that R_X3_ > R_X1_ > R_X2_ > R_X5_ > R_X4_; namely, the effect on acetoin production occurs in the following order: induction temperature > initial OD_600_ > initial pH > induction time > shaking speed. The optimal induction conditions were an initial OD_600_ of 0.6, initial pH of 7.5, induction temperature of 37 °C, and static induction for 8 h, and the maximum acetoin production was 106.93 mg/L.

### 3.6. Fermentation/Whole-Cell Bioconversion of Soybean with Recombinant L. lactis

Considering that the optimal induction temperature was 37 °C in the orthogonal test, and that a relatively high temperature is commonly used in the industry to make fermented products, 37 °C and 40 °C were set to explore the optimal fermentation temperature of recombinant bacteria with soybean as the substrate. As shown in Figure 6, except for the soybean control, the acetoin produced at 37 °C was generally higher than that at 40 °C, and the largest difference (18.22 mg/L) was observed in FS2. It may be that a high temperature affects the enzyme activity of ALDC, which was also confirmed in the above induction condition experiment (Figure 5e). At the same fermentation temperature, FS2 was observed to generate the highest production of acetoin. In comparison with FS1 and FS3, nisin contained in both phases Ⅰ and Ⅱ induced ALDC production throughout the whole-cell fermentation of acetoin biosynthesis, thereby increasing the production of acetoin in fermented soybeans. In addition, a relatively high level of acetoin was detected in FS1, although nisin was not added in phase II. This may be attributed to the residual nisin in the cytoplasm, and/or lactose and galactose in the fermented soybeans as inducers. Although the concentration of acetoin in our study was lower than the high-level acetoin production (5.8 g/L) in metabolically engineered *L. lactis* strain CS4701m, which was constructed by inactivating the all-competing product pathway and expressing one diacetyl reductase [23], we combined raw soybean and whole-cell bioconversion to produce 79.43 mg/L acetoin by using an *aldC*-heterologous expression recombinant *L. lactis*, which laid the theoretical foundation for adding special flavors to fermented soybean products.

## 4. Conclusions

In this study, the effect of *aldC* gene overexpression on acetoin production in *L. lactis* was elucidated, after which the production was further increased by optimizing the induction conditions. In addition, the whole cells of NZ9000/pNZ8048-*aldC* were employed as the catalyst, and a fermented soybean produced 79.43 mg/L acetoin. Our results provide a solid basis for future research involving the *aldC* gene in the production of acetoin through whole-cell bioconversion.

## Figures and Tables

**Figure 1 foods-12-01317-f001:**
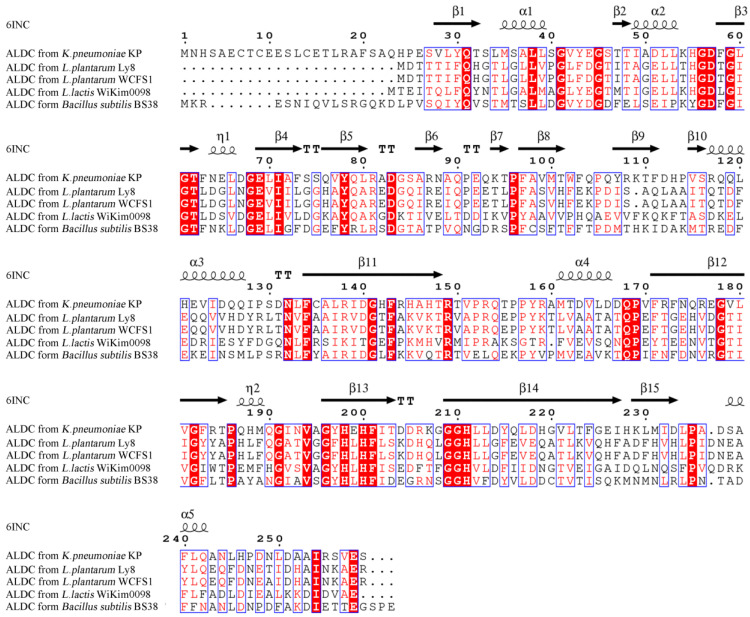
Sequence alignment of ALDC from *K. pneumonia* KP, *L. plantarum* Ly8, *L. plantarum* WCFS1, *L. lactis* WiKim0098 and *Bacillus subtilis* BS38. Residues conserved in all five proteins are highlighted with a red background. ALDC from *K. pneumoniae* secondary structure elements (6INC) are indicated at the top of each panel. α: α-helix; β: β-sheet; T: β-turns; η: 3/10-helix.

**Figure 2 foods-12-01317-f002:**
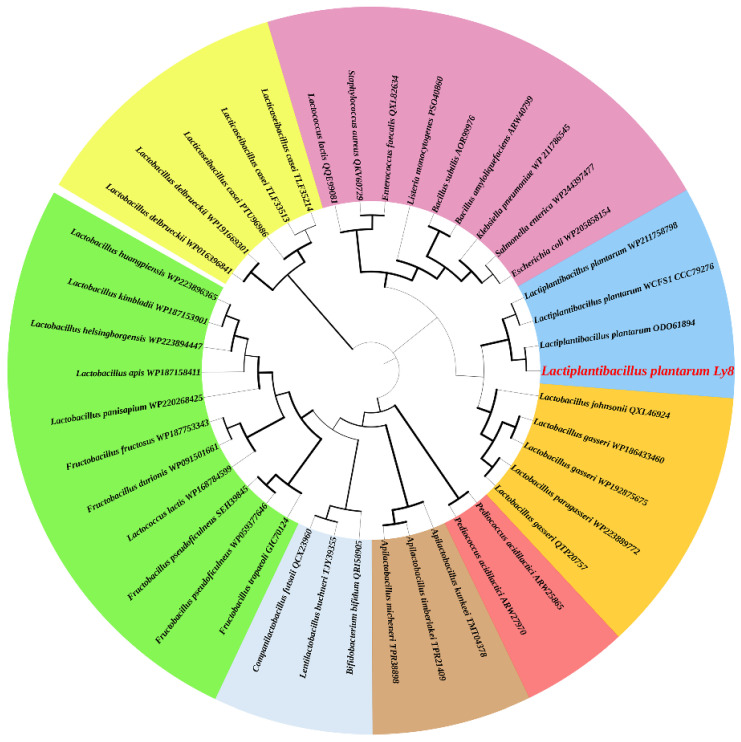
A phylogenetic maximum likelihood tree showing the relationships between ALDC from *L. plantarum* Ly8 and other bacterial ALDC proteins. The thickness of the branches indicates the percentages at which the given branches were supported with 1000 bootstrap replications. The GenBank accession numbers are displayed after the species name. *L. plantarum* Ly8 is marked with red font.

**Figure 3 foods-12-01317-f003:**
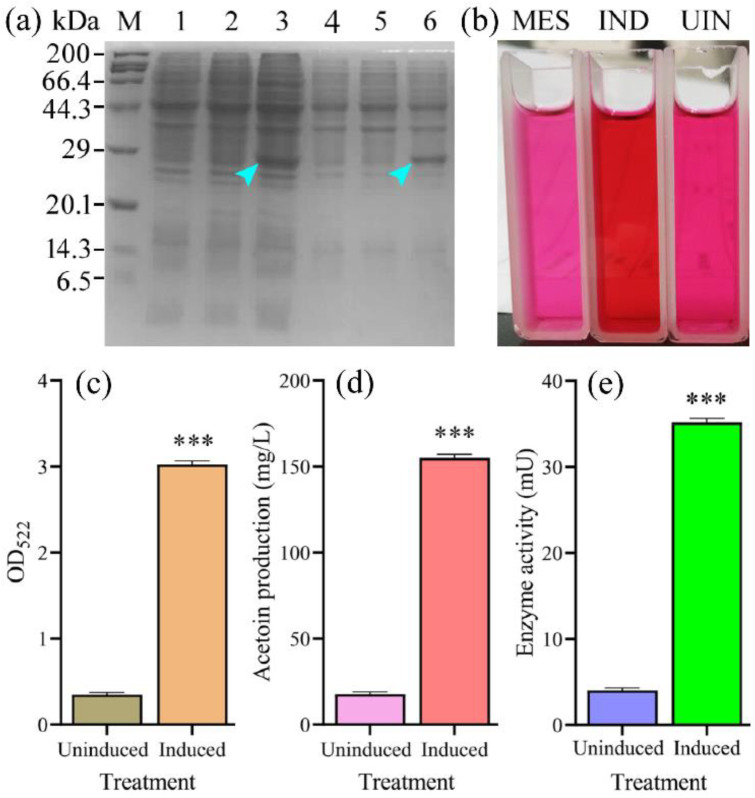
SDS-PAGE and enzymatic analyses of recombinant ALDC. (**a**) SDS-PAGE analysis. M, protein molecular weight marker; Lanes 1–3, proteins from cell pellet; Lanes 4–6, proteins from supernatant; Lanes 1 and 4, proteins from NZ9000/pNZ8048; Lanes 2 and 5, proteins from NZ9000/pNZ8048-*aldC* without nisin induction; Lanes 3 and 6, proteins from NZ9000/pNZ8048-*aldC* with nisin induction; arrows, target protein bands. (**b**) The result of the chromogenic reaction. MES, MES buffer used for zeroing; IND, NZ9000/pNZ8048-*aldC* with nisin induction; UIN: NZ9000/pNZ8048-*aldC* without nisin induction. (**c**) The OD_522_ of NZ9000/pNZ8048-*aldC* with or without nisin induction. (**d**) The acetoin production of NZ9000/pNZ8048-*aldC* with or without nisin induction. (**e**) The enzyme activity of NZ9000/pNZ8048-*aldC* with or without nisin induction. Data are presented as the mean ± SD. *** indicates a significant difference (*p* < 0.001).

**Figure 4 foods-12-01317-f004:**
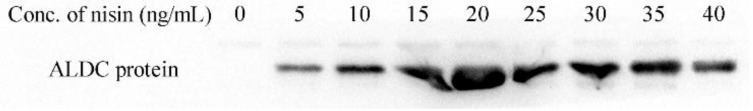
Effect of nisin concentrations on ALDC expression.

**Figure 5 foods-12-01317-f005:**
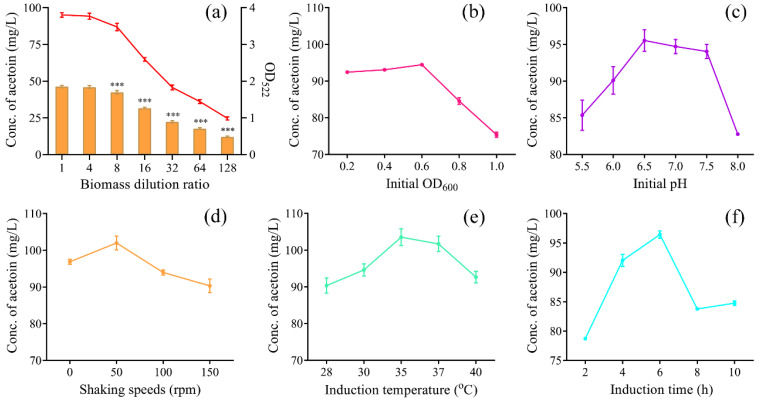
The effect of biomass dilution (**a**) and induction conditions (**b**–**f**) on acetoin production. *** denotes a significant difference at *p* < 0.001.

**Figure 6 foods-12-01317-f006:**
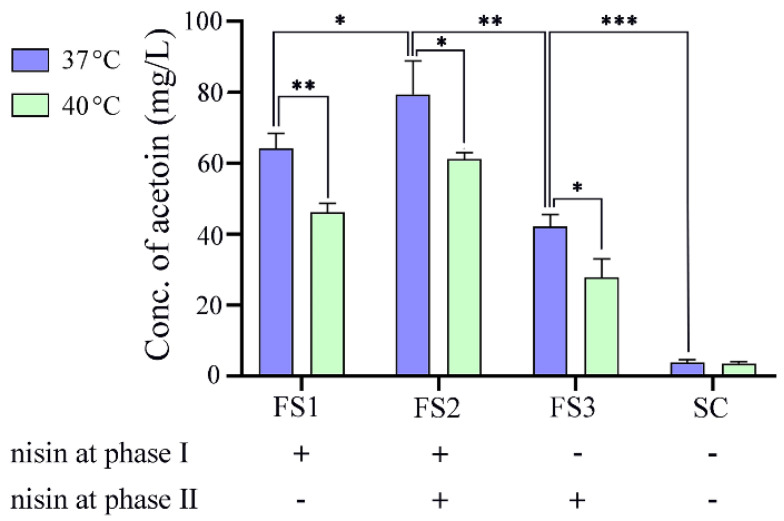
Acetoin production in NZ9000/pNZ8048-*aldC* fermented soybeans. + and − indicate with or without nisin induction at phase I and II. SC is short for soybean control. The error bars show standard deviations from three repeated experiments, and significant differences are presented as * (*p* < 0.05), ** (*p* < 0.01) and *** (*p* < 0.001).

**Table 1 foods-12-01317-t001:** Bacterial strains, plasmids and primers used in this study.

Strain, Plasmid and Primer		Relevant Characteristic or Sequence (5′-3′) ^a^	Source
Strains	*L*. *plantarum* Ly8	Wild-type strain, isolated from Douchi	This study
	*L. lactis* NZ9000	*L. lactis* MG1363 *pepN::nisRK*	Biovector
	*L. lactis* NZ9000/pNZ8048	*L. lactis* MG1363 *pepN::nisRK*, containing plasmid pNZ8048	This study
	*L. lactis* NZ9000/pNZ8048-*aldC*	*L. lactis* MG1363 *pepN::nisRK*, containing plasmid pNZ8048-*aldC*	This study
	*E. coli* DH5α	F^-^, endA1, glnV44, thi-1, recA1, relA1, gyrA96, deoR, nupG, Φ80dlacZ ΔM15, Δ(lacZYA-argF) U169, hsdR17(rK−, mK+), λ−	Takara
Plasmids	pNZ8048	Lactic acid bacteria expression vector, CmR	Biovector
	pNZ8048-*aldC*	Lactic acid bacteria expression vector, CmR, containing *aldC* gene	This study
Primers	*aldC*-F	CAGCTGCAGACATGGACACAACAACAATTTT	This study
	*aldC*-R	CTTCTAGATTAGTGGTGGTGGTGGTGGTGACGTTCAGCTTTATTAATGG	This study

^a^ The underlined nucleotides indicate restriction enzyme sites for *Pst* I (CTGCAG) or *Xba* I (TCTAGA).

**Table 2 foods-12-01317-t002:** Factors and levels for the orthogonal test to optimize the induction conditions.

Factor	Item	Level		
		1	2	3
X_1_	Initial OD_600_	0.4	0.6	0.8
X_2_	Initial pH	6.5	7.0	7.5
X_3_	Induction temperature (°C)	30	35	37
X_4_	Shaking speed (rpm)	0	50	100
X_5_	Induction time (h)	4	6	8

**Table 3 foods-12-01317-t003:** The L_18_ (3^5^) orthogonal test applied for optimizing the production of acetoin ^a^.

No.	Factor					OD_522_	Acetoin (mg/L)
	X_1_ (OD_600_)	X_2_ (pH)	X_3_ (°C)	X_4_ (rpm)	X_5_ (h)		
1	0.4	6.5	30	0	4	1.61 ± 0.03	82.18 ± 1.71
2	0.6	7.0	30	50	6	1.49 ± 0.02	76.01 ± 1.25
3	0.8	7.5	30	100	8	0.72 ± 0.01	36.91 ± 0.71
4	0.6	7.0	35	0	4	0.84 ± 0.03	42.90 ± 1.42
5	0.8	7.5	35	50	6	1.44 ± 0.01	73.67 ± 0.37
6	0.4	6.5	35	100	8	1.61 ± 0.01	82.40 ± 0.54
7	0.4	7.5	37	0	6	1.68 ± 0.04	86.21 ± 2.13
8	0.6	6.5	37	50	8	1.45 ± 0.03	74.03 ± 1.54
9	0.8	7.0	37	100	4	2.01 ± 0.04	102.74 ± 1.95
10	0.8	7.0	30	0	8	1.46 ± 0.03	74.49 ± 1.76
11	0.4	7.5	30	50	4	1.64 ± 0.03	83.80 ± 1.59
12	0.6	6.5	30	100	6	1.28 ± 0.02	65.56 ± 1.26
13	0.8	6.5	35	0	6	1.24 ± 0.11	63.36 ± 5.86
14	0.4	7.0	35	50	8	1.75 ± 0.04	89.75 ± 2.21
15	0.6	7.5	35	100	4	1.76 ± 0.01	89.91 ± 0.41
16	0.6	7.5	37	0	8	2.09 ± 0.02	106.93 ± 1.16
17	0.8	6.5	37	50	4	1.42 ± 0.03	72.90 ± 1.32
18	0.4	7.0	37	100	6	1.76 ± 0.03	90.28 ± 1.32
K_1_	85.78	73.40	69.83	76.02	79.08		
K_2_	75.90	79.37	73.67	78.37	75.86		
K_3_	70.68	79.58	88.86	77.96	77.42		
R	15.1	6.18	19.03	2.35	3.22		

^a^ K = sum of the experimental indices of this level in each factor; R = difference between the average of K_max_ and K_min_ values.

## Data Availability

Data are contained within the article.

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
