# Peer review of "Whole-Cell Biocatalytic Production of Acetoin with an aldC-Overexpressing Lactococcus lactis Using Soybean as Substrate"

_foods, 2023, doi:10.3390/foods12061317_

Round 1
Reviewer 1 Report
In my opinion, the manuscript needs some revision. In particular, the following changes should be considered by the authors.
The authors should discuss the advantages and drawbacks on the nisin-inducible system.Why did the authors select this induction system?
What the authors comment about the serial re-pitching on the acetoin production by La. lactis
Author Response
Comments and Suggestions for Authors
In my opinion, the manuscript needs some revision. In particular, the following changes should be considered by the authors.
- The authors should discuss the advantages and drawbacks on the nisin-inducible system. Why did the authors select this induction system?
Answer:
The discussion about nisin-inducible system has been added in the Introduction (L56-58). Please see the yellow shading mark.
- What the authors comment about the serial re-pitching on the acetoin production by La. lactis
Answer:
We don't understand what the reviewer means. Figure 5 ? We have described the situation of acetoin production in detail.
Reviewer 2 Report
Please check some typing mistakes throughout the manuscript.
i.e.:
· Abstract - Lactoplantbacillus plantarum needs to write in italic. Check all manuscript. Additionally, check the paper, white showed the correct form of the Lactoplantbacillus and Lactococcus abbreviation (https://doi.org/10.1099/ijsem.0.004107)
· Introduction- Molecular formula from acetoin C4H8O2 needs to write in subscribed numbers.
· Check all manuscript.
· L240- in vitro needs to write in italic.
· L34-35 - Rewrite the sentence describing the full names first of the government agencies and then the abbreviations.
· L185 and L301- an L18 (35) or an L18 (35)
Material and methods
If acetoin is naturally produced by strains of L. plantarum, why was the gene inserted into L. lactis?
Why is the L. plantarum Ly8 described as a putative probiotic organism? Is there any scientific evidence of these probiotic effects? Since not all L. plantarum shows beneficial effects on the health of the hosts, not being considered a probiotic.
How were the SDS-PAGE analyses performed? Describe the running conditions, sample processing, and acrylamide concentration of the gels. This information may be included in the supplementary material.
It's not clear why the nisin was used to induce the production of ALDC. Has any previous study reported it?
Why were the temperatures of 37 ºC and 40 ºC used in the ALDC production tests since in the orthogonal tests the temperature of 40 ºC was not evaluated?
L-199-205- Rewrite the sentence. The description of the treatment is very confusing.
What statistical software was employed for statistical tests?
Results and discussion
Why was the temperature of 40 ºC not used in the orthogonal tests? Since this temperature was used in the fermentation tests with the recombinant L. lactis.
The results demonstrate that the temperature of 40 ºC showed a significantly less production of acetoin by the recombinant strain. Could it be that the other conditions tested during the orthogonal tests could not favor a greater production of acetoin at 40 ºC?
If soy does not have the carbohydrates lactose and galactose in its carbohydrate composition, which compounds present in soy could have promoted this ALDC production in soy fermented products without the presence of nisin?
Include in the discussion compounds present in the soybean food matrix that could promote this acetoin production in the treatments without nisin.
Describe each treatment in Figure 6. What is SC? It is not clear.
Author Response
Comments and Suggestions for Authors
Please check some typing mistakes throughout the manuscript.
i.e.:
- Abstract - Lactoplantbacillus plantarumneeds to write in italic. Check all manuscript. Additionally, check the paper, white showed the correct form of the Lactoplantbacillusand
Lactococcus abbreviation (https://doi.org/10.1099/ijsem.0.004107).
Answer:
In the revised version, the Lactoplantbacillus and Lactococcus abbreviation have been changed to L. plantarum and L. lactis. And all strain names have been italicized. Please see the yellow shading mark. In addition, the strains’ names in Figure 1 were also changed.
- Introduction- Molecular formula from acetoin C4H8O2needs to write in subscribed numbers.
Answer:
The subscribed numbers of acetoin have been added to the revised version (L31), please see the yellow shading mark.
- Check all manuscript.
Answer:
Done. we have checked all manuscript.
- L240- in vitroneeds to write in italic.
Answer:
Done. in vitro have been italicized in revised version (L243).
- L34-35 - Rewrite the sentence describing the full names first of the government agencies and then the abbreviations.
Answer:
The government describing was rewritten (L34-35), please see the yellow shading mark.
- L185 and L301- an L18 (35) or an L18 (35)
Answer:
The correct way to write an orthogonal table is L18 (35), it has been modified in revised version (L186, L304, L311), please see the yellow shading mark.
Material and methods
- If acetoin is naturally produced by strains of L. plantarum, why was the gene inserted into L. lactis?
Answer:
As a GRAS microorganism, L. lactis was widely used in the food industry. L. lactis NZ9000 has been used as a superior host for recombinant protein expression, this strain would be a suitable host for ALDC production. Recent advances in the molecular characterization of L. lactis and the development of L. lactis-compatible genetic engineering tools have increased the versatility of this organism as a means of protein production.
- Why is the L. plantarumLy8 described as a putative probiotic organism? Is there any scientific evidence of these probiotic effects? Since not all L. plantarumshows beneficial effects on the health of the hosts, not being considered a probiotic.
Answer:
- plantarumLy8 was isolated from food in our previous experiments, it has strong acid and bile salt resistance and can effectively regulate the balance of intestinal flora, that means Ly8 has the prerequisites to be a putative probiotic organism. To avoid disagreement, we have deleted the words "putative probiotic organism".
- How were the SDS-PAGE analyses performed? Describe the running conditions, sample processing, and acrylamide concentration of the gels. This information may be included in the supplementary material.
Answer:
We conducted SDS-PAGE according to the methods reported in a literature (https://doi.org/10.1016/j.micpath.2017.07.039), and added a sentence in the revised version (L134-135).
- It's not clear why the nisin was used to induce the production of ALDC. Has any previous study reported it?
Answer:
aldC gene was cloned into the pNZ8048 vector, which has a PnisA promoter. The PnisA promoter need nisin to induce. To understand clearly, the nisin-inducible system was described in the Introduction (L56-58). Please see the yellow shading mark.
- Why were the temperatures of 37 ºC and 40 ºC used in the ALDC production tests since in the orthogonal tests the temperature of 40 ºC was not evaluated?
Answer:
In the whole-cell bioconversion experiment, we want to compare the optimal temperature obtained by orthogonal test (37 ºC) with the temperature commonly used in industrial fermentation on acetoin production in soybeans.
- L-199-205- Rewrite the sentence. The description of the treatment is very confusing.
Answer:
The sentences were rewritten in the revised version (L200-207), please see the yellow shading mark).
- What statistical software was employed for statistical tests?
Answer:
The software we employed for statistical tests was SPSS 22. The sentence “All statistical analyses were conducted using SPSS 22 (IBM, Armonk, NY, USA)” was added in the manuscript (L211-212).
Results and discussion
- Why was the temperature of 40 ºC not used in the orthogonal tests? Since this temperature was used in the fermentation tests with the recombinant L. lactis.
Answer:
In the temperature optimization experiment, the yield of acetoin was from low to high and then to low. The highest production of acetoin occurs at 35 ºC. 30 ºC and 37 ºC are the two nearby temperatures, so 30 ºC, 35 ºC and 37 ºC are selected for the orthogonal experiment.
- The results demonstrate that the temperature of 40 ºC showed a significantly less production of acetoin by the recombinant strain. Could it be that the other conditions tested during the orthogonal tests could not favor a greater production of acetoin at 40 ºC?
Answer:
In the orthogonal experiment, we did not perform the temperature of 40 ºC. Based on the results of single factor experiments and soybean fermentation by recombinant bacteria, 40 ºC may be unfavorable for the production of acetoin.
- If soy does not have the carbohydrates lactose and galactose in its carbohydrate composition, which compounds present in soy could have promoted this ALDC production in soy fermented products without the presence of nisin?
Answer:
We don’t know. Lactose and galactose in the fermented soybeans may be the inducer. A sentence that “In addition, a relatively high level of acetoin was detected in FS1, although nisin was not added in phase II. This may be attributed to the residual nisin in the cytoplasm and/or lactose and galactose in the fermented soybeans as inducers” was added.
- Include in the discussion compounds present in the soybean food matrix that could promote this acetoin production in the treatments without nisin.
Answer:
The sentence “In addition, a relatively high level of acetoin was detected in FS1, although nisin was not added in phase II. This may be attributed to the residual nisin in the cytoplasm and/or lactose and galactose in the fermented soybeans as inducers” was added (L325-327).
- Describe each treatment in Figure 6. What is SC? It is not clear.
Answer:
Done. Please L336 in the revised manuscript.
Reviewer 3 Report
in this study, the authors the aldC gene was cloned from Lactiplantibacillus plantarum and overexpressed in Lactococcus lactis. the expression of aldC gene was analyzed at different conditions for acetoin production. Subsequently soybeans were used as substrate for production of acetoin. The research is interesting for the application of probiotic strains for the whole cell production of acetoin using modified GRAS strains. The manuscript is clear and the used methodology is well described.
The English language needs minor editing.
The conditions for Acetoin production were studied and optimized. Then Acetoin was produced using soybean as substrate.
Please check the use of italics for the name of organisms.
Specific
Title: Please consider using "....soybean as substrate" instead of "...soybean as substance".
The amount of acetoin produced(79 mg/L), how does it relate with industrial production o acetoin, or the amount is sufficient to change functional or flavor properties of the fermented product? is there any application possibility for the industrial acetoin production from soybean for flavor and aroma agents?
The optimal production temperature of 37°C is related to the strain, or the enzyme performance?
Author Response
Comments and Suggestions for Authors
in this study, the authors the aldC gene was cloned from Lactiplantibacillus plantarum and overexpressed in Lactococcus lactis. the expression of aldC gene was analyzed at different conditions for acetoin production. Subsequently soybeans were used as substrate for production of acetoin. The research is interesting for the application of probiotic strains for the whole cell production of acetoin using modified GRAS strains. The manuscript is clear and the used methodology is well described.
The English language needs minor editing.
- The conditions for Acetoin production were studied and optimized. Then Acetoin was produced using soybean as substrate.
Please check the use of italics for the name of organisms.
Answer:
All the name of organisms and gene have been corrected in revised version.
- Specific
Title: Please consider using "....soybean as substrate" instead of "...soybean as substance".
Answer:
The word "substance" in title has been replaced by the "substrate".
- The amount of acetoin produced (79 mg/L), how does it relate with industrial production o acetoin, or the amount is sufficient to change functional or flavor properties of the fermented product? is there any application possibility for the industrial acetoin production from soybean for flavor and aroma agents?
Answer:
In this study, the acetoin production from soybeans was much lower than that from industrial production, but the flavor of fermented soybeans with NZ9000/pNZ8048-aldC was significantly changed and the edible taste was increased.
- The optimal production temperature of 37°C is related to the strain, or the enzyme performance?
Answer:
The aldC gene was cloned form L. plantarum and over-expression in L. lactis. The optimum growth temperature of these two strains was 37 °C. It is reasonable that the ALDC enzyme performance at 37°C is optimal.
Round 2
Reviewer 2 Report
The changes made by the authors were satisfactory and the questions were answered. I believe that the present manuscript contributes to a viable biological alternative for the production of acetoin. Although it is not the focus of this manuscript, I hope that furthermore studies on a viable way to purification of this biomolecule will be presented, since food matrices present a large number of compounds that make it difficult and increase the cost of isolation and purification of biomolecules.